# Advances in Learning Bayesian Networks of Bounded Treewidth

**Siqi Nie**
Rensselaer Polytechnic Institute
Troy, NY, USA
nies@rpi.edu

**Denis D. Mauá**
University of São Paulo
São Paulo, Brazil
denis.maua@usp.br

**Cassio P. de Campos**
Queen's University Belfast
Belfast, UK
c.decampos@qub.ac.uk

**Qiang Ji**
Rensselaer Polytechnic Institute
Troy, NY, USA
qji@ecse.rpi.edu

## Abstract

This work presents novel algorithms for learning Bayesian networks of bounded treewidth. Both exact and approximate methods are developed. The exact method combines mixed integer linear programming formulations for structure learning and treewidth computation. The approximate method consists in sampling $k$-trees (maximal graphs of treewidth $k$), and subsequently selecting, exactly or approximately, the best structure whose moral graph is a subgraph of that $k$-tree. The approaches are empirically compared to each other and to state-of-the-art methods on a collection of public data sets with up to 100 variables.

## 1 Introduction

Bayesian networks are graphical models widely used to represent joint probability distributions on complex multivariate domains. A Bayesian network comprises two parts: a directed acyclic graph (the structure) describing the relationships among the variables in the model, and a collection of conditional probability tables from which the joint distribution can be reconstructed. As the number of variables in the model increases, specifying the underlying structure becomes a daunting task, and practitioners often resort to learning Bayesian networks directly from data. Here, learning a Bayesian network refers to inferring its structure from data, a task known to be NP-hard [9].

Learned Bayesian networks are commonly used for drawing inferences such as querying the posterior probability of some variable given some evidence or finding the mode of the posterior joint distribution. Those inferences are NP-hard to compute even approximately [23], and all known exact and provably good algorithms have worst-case time complexity exponential in the treewidth, which is a measure of the tree-likeness of the structure. In fact, under widely believed assumptions from complexity theory, exponential time complexity in the treewidth is inevitable for any algorithm that performs exact inference [7, 20]. Thus, learning networks of small treewidth is essential if one wishes to ensure reliable and efficient inference. This is particularly important in the presence of missing data, when learning becomes intertwined with inference [16]. There is a second reason to limit the treewidth. Previous empirical results [15, 22] suggest that bounding the treewidth improves model performance on unseen data, hence improving the model generalization ability.

In this paper we present two novel ideas for score-based Bayesian network learning with a hard constraint on treewidth. The first one is a mixed-integer linear programming (MILP) formulation of the problem (Section 3) that builds on existing MILP formulations for unconstrained learning of Bayesian networks [10, 11] and for computing the treewidth of a graph [17]. Unlike the MILP

formulation of Parviainen et al. [21], the MILP problem we generate is of polynomial size in the number of variables, and dispense with the use of cutting planes techniques. This makes for a clean and succinct formulation that can be solved with a single call of any MILP optimizer. We provide some empirical evidence (in Section 5) that suggests that our approach is not only simpler but often faster. It also outperforms the dynamic programming approach of Korhonen and Parviainen [19].

Since linear programming relaxations are used for solving the MILP problem, any MILP formulation can be used to provide approximate solutions and error estimates in an anytime fashion (i.e., the method can be stopped at any time during the computation with a feasible solution whose quality monotonically improves with time). However, the MILP formulations (both ours and that of Parviainen et al. [21]) cannot cope with very large domains, even if we settle for approximate solutions. In order to deal with large domains, we devise (in Section 4) an approximate method based on a uniform sampling of $k$-trees (maximal chordal graphs of treewidth $k$), which is achieved by using a fast computable bijection between $k$-trees and Dandelion codes [6]. For each sampled $k$-tree, we either run an exact algorithm similar to the one in [19] (when computationally appealing) to learn the score-maximizing network whose moral graph is a subgraph of that $k$-tree, or we resort to a more efficient method that takes partial variable orderings uniformly at random from a (relatively small) space of orderings that are compatible with the $k$-tree. We show empirically (in Section 5) that our sampling-based methods are very effective in learning close to optimal structures and scale up to large domains. We conclude in Section 6 and point out possible future work. We begin with some background knowledge and literature review on learning Bayesian networks (Section 2).

## 2    Bayesian Network Structure Learning

Let $N$ be $\{1, \ldots, n\}$ and consider a finite set $X = \{X_i : i \in N\}$ of categorical random variables $X_i$ taking values in finite sets $\mathcal{X}_i$. A Bayesian network is a triple $(X, G, \theta)$, where $G = (N, A)$ is a directed acyclic graph (DAG) whose nodes are in one-to-one correspondence with variables in $X$, and $\theta = \{\theta_i(x_i, x_{G_i})\}$ is a set of numerical parameters specifying (conditional) probabilities $\theta_i(x_i, x_{G_i}) = \Pr(x_i|x_{G_i})$, for every node $i$ in $G$, value $x_i$ of $X_i$ and assignment $x_{G_i}$ to the parents $G_i$ of $X_i$ in $G$. The structure $G$ of the network represents a set of stochastic independence assessments among variables in $X$ called graphical Markov conditions: every variable $X_i$ is conditionally independent of its nondescendant nonparents given its parents. As a consequence, a Bayesian network uniquely defines a joint probability distribution over $X$ as the product of its parameters.

As it is common in the literature, we formulate the problem of Bayesian network learning as an optimization over DAG structures guided by a *score function*. We only require that (i) the score function can be written as a sum of local score functions $s_i(G_i)$, $i \in N$, each depending only on the corresponding parent set $G_i$ and on the data, and (ii) the local score functions can be efficiently computed and stored [13, 14]. These properties are satisfied by commonly used score functions such as the Bayesian Dirichlet equivalent uniform score [18]. We assume the reader is familiar with graph-theoretic concepts such as polytrees, chordal graphs, chordalizations, moral graphs, moralizations, topological orders, (perfect) elimination orders, fill-in edges and clique-trees. References [1] and [20] are good starting points to the topic.

Most score functions penalize model complexity in order to avoid overfitting. The way scores penalize model complexity generally leads to learning structures of bounded in-degree, but even bounded in-degree graphs can have high treewidth (for instance, directed square grids have treewidth equal to the square root of the number of nodes, yet have maximum in-degree equal to two), which brings difficulty to subsequent probabilistic inferences with the model [5].

The goal of this work is to develop methods that search for

$$G^* = \underset{G \in \mathcal{G}_{N,k}}{\operatorname{argmax}} \sum_{i \in N} s_i(G_i) \,, \tag{1}$$

where $\mathcal{G}_{N,k}$ is the set of all DAGs with node set $N$ and treewidth at most $k$. Dasgupta proved NP-hardness of learning polytrees of bounded treewidth when the score is data log likelihood [12]. Korhonen and Parviainen [19] adapted Srebro's complexity result for Markov networks [25] to show that learning Bayesian networks of treewidth two or greater is NP-hard.

In comparison to the unconstrained problem, few algorithms have been designed for the bounded treewidth case. Korhonen and Parviainen [19] developed an exact algorithm based on dynamic

programming that learns optimal $n$-node structures of treewidth at most $w$ in time $3^n n^{w+O(1)}$, which is above the $2^n n^{O(1)}$ time required by the best worst-case algorithms for learning optimal Bayesian networks with no constraint on treewidth [24]. We shall refer to their method in the rest of this paper as K&P (after the authors' initials). Elidan and Gould [15] combined several heuristics to treewidth computation and network structure learning in order to design approximate methods. Others have addressed the similar (but not equivalent) problem of learning undirected models of bounded treewidth [2, 8, 25]. Very recently, there seems to be an increase of interest in the topic. Berg et al. [4] showed that the problem of learning bounded treewidth Bayesian networks can be reduced to a weighted maximum satisfiability problem, and subsequently solved by weighted MAX-SAT solvers. They report experimental results showing that their approach outperforms K&P. In the same year, Parviainen et al. [21] showed that the problem can be reduced to a MILP. Their reduced MILP problem however has exponentially many constraints in the number of variables. Following the work of Cussens [10], the authors avoid creating such large programs by a cutting plane generation mechanism, which iteratively includes a new constraint while the optimum is not found. The generation of each new constraint (cutting plane) requires solving another MILP problem. We shall refer to their method from now on as TWILP (after the name of the software package the authors provide).

## 3 A Mixed Integer Linear Programming Approach

The first contribution of this work is the MILP formulation that we design to solve the problem of structure learning with bounded treewidth. MILP formulations have shown to be very effective for learning Bayesian networks with no constraint on treewidth [3, 10], surpassing other attempts in a range of data sets. The formulation is based on combining the MILP formulation for structure learning in [11] with the MILP formulation presented in [17] for computing the treewidth of an undirected graph. There are however notable differences: for instance, we do not enforce a linear elimination ordering of nodes; instead we allow for partial orders which capture the equivalence between different orders in terms of minimizing treewidth, and we represent such partial order by real numbers instead of integers. We avoid the use of sophisticate techniques for solving MILP problems such as constraint generation [3, 10], which allows for an easy implementation and parallelization (MILP optimizers such as CPLEX can take advantage of that).

For each node $i$ in $N$, let $\mathcal{P}_i$ be the collection of allowed parent sets for that node (these sets can be specified manually by the user or simply defined as the subsets of $N \setminus \{i\}$ with cardinality less than a given bound). We denote an element of $\mathcal{P}_i$ as $P_{it}$, with $t = 1, \dots, r_i$ and $r_i = |\mathcal{P}_i|$ (hence $P_{it} \subset N$). We will refer to a DAG as valid if its node set is $N$ and the parent set of each node $i$ in it is an element of $\mathcal{P}_i$. The following MILP problem can be used to find valid DAGs whose treewidth is at most $w$:

$$\text{Maximize} \quad \sum_{it} p_{it} \cdot s_i(P_{it}) \quad \text{subject to} \tag{2}$$

$$\sum_{j \in N} y_{ij} \leq w, \qquad \forall i \in N, \tag{3a}$$

$$(n+1) \cdot y_{ij} \leq n + z_j - z_i, \qquad \forall i, j \in N, \tag{3b}$$

$$y_{ij} + y_{ik} - y_{jk} - y_{kj} \leq 1, \qquad \forall i, j, k \in N, \tag{3c}$$

$$\sum_t p_{it} = 1, \qquad \forall i \in N, \tag{4a}$$

$$(n+1)p_{it} \leq n + v_j - v_i, \qquad \forall i \in N, \forall t \in \{1, \dots, r_i\}, \forall j \in P_{it}, \tag{4b}$$

$$p_{it} \leq y_{ij} + y_{ji}, \qquad \forall i \in N, \forall t \in \{1, \dots, r_i\}, \forall j \in P_{it}, \tag{4c}$$

$$p_{it} \leq y_{jk} + y_{kj}, \qquad \forall i \in N, \forall t \in \{1, \dots, r_i\}, \forall j, k \in P_{it}, \tag{4d}$$

$$z_i \in [0, n], \; v_i \in [0, n], \; y_{ij} \in \{0, 1\}, \; p_{it} \in \{0, 1\} \qquad \forall i, j \in N, \forall t \in \{1, \dots, r_i\}. \tag{5}$$

The variables $p_{it}$ define which parent sets are chosen, while the variables $v_i$ guarantee that those choices respect a linear ordering of the variables, and hence that the corresponding directed graph is acyclic. The variables $y_{ij}$ specify a chordal moralization of this DAG with arcs respecting an elimination ordering of width at most $w$, which is given by the variables $z_i$.

The following result shows that any solution to the MILP above can be decoded into a chordal graph of bounded treewidth and a suitable perfect elimination ordering.

**Lemma 1.** *Let $z_i, y_{ij}, i, j \in N$, be variables satisfying Constraints* (4) *and* (5). *Then the undirected graph $M = (N, E)$, where $E = \{ij \in N \times N : y_{ij} = 1 \text{ or } y_{ji} = 1\}$, is chordal and has treewidth at most $w$. Any elimination ordering that extends the weak ordering induced by $z_i$ is perfect for $M$.*

The graph $M$ is used in the formulation as a template for the moral graph of a valid DAG:

**Lemma 2.** *Let $v_i, p_{it}, i \in N, t = 1, \ldots, r_i$, be variables satisfying Constraints* (4) *and* (5). *Then the directed graph $G = (N, A)$, where $G_i = \{j : p_{it} = 1 \text{ and } j \in P_{it}\}$, is acyclic and valid. Moreover the moral graph of $G$ is a subgraph of the graph $M$ defined in the previous lemma.*

The previous lemmas suffice to show that the solutions of the MILP problem can be decoded into valid DAGs of bounded treewidth:

**Theorem 1.** *Any solution to the MILP can be decoded into a valid DAG of treewidth less than or equal to $w$. In particular, the decoding of an optimal solution solves* (1).

The MILP formulation can be directly fed into any off-the-shelf MILP optimizer. Most MILP optimizers (e.g. CPLEX) can be prematurely stopped while providing an incumbent solution and an error estimate. Moreover, given enough resources (time and memory), these solvers always find optimal solutions. Hence, the MILP formulation provides an anytime algorithm that can be used to provide both exact and approximate solutions.

The bottleneck in terms of efficiency of the MILP construction lies in the specification of Constraints (3c) and (4d), as there are $\Theta(n^3)$ such constraints. Thus, as $n$ increases even the linear relaxations of the MILP problem become hard to solve. We demonstrate empirically in Section 5 that the quality of solutions found by the MILP approach in a reasonable amount of time degrades quickly as the number of variables exceeds a few dozens. In the next section, we present an approximate algorithm to overcome such limitations and handle large domains.

## 4 A Sampling Based Approach

A successful method for learning Bayesian networks of unconstrained treewidth on large domains is order-based local search, which consists in sampling topological orderings for the variables and selecting optimal compatible DAGs [26]. Given a topological ordering, the optimal DAG can be found in linear time (assuming scores are given as input), hence rendering order-based search really effective in exploring the solution space. A naive extension of that approach to the bounded treewidth case would be to (i) sample a topological order, (ii) find the optimal compatible DAG, (iii) verify the treewidth and discard if it exceeds the desired bound. There are two serious issues with that approach. First, verifying the treewidth is an NP-hard problem, and even if there are linear-time algorithms (which are exponential in the treewidth), they perform poorly in practice; second, the vast majority of structures would be discarded, since the most used score functions penalize the number of free parameters, which correlates poorly with treewidth [5].

In this section, we propose a more sophisticate extension of order-based search to learn bounded treewidth structures. Our method relies on sampling $k$-trees, which are defined inductively as follows [6]. A complete graph with $k + 1$ nodes (i.e., a $(k + 1)$-clique) is a $k$-tree. Let $T_k = (V, E)$ be a $k$-tree, $K$ be a $k$-clique in it, and $v$ be a node not in $V$. Then the graph obtained by connecting $v$ to every node in $K$ is also a $k$-tree. A $k$-tree is a maximal graph of treewidth $k$ in the sense that no edge can be added without increasing the treewidth. Every graph of treewidth at most $k$ is a subgraph of some $k$-tree. Hence, Bayesian networks of treewidth bounded by $k$ are exactly those whose moral graph is a subgraph of some $k$-tree [19]. We are interested in $k$-trees over the nodes $N$ of the Bayesian network and where $k = w$ is the bound we impose to the treewidth.

Caminiti et al. [6] proposed a linear time method (in both $n$ and $k$) for coding and decoding $k$-trees into what is called *(generalized) Dandelion codes*. They also established a bijection between Dandelion codes and $k$-trees. Hence, sampling Dandelion codes is essentially equivalent to sampling $k$-trees. The former however is computationally much easier and faster to perform, especially if we want to draw samples uniformly at random (uniform sampling provides good coverage of the space and produces low variance estimates across data sets). Formally, a Dandelion code is a pair $(Q, S)$, where $Q \subseteq N$ with $|Q| = k$ and $S$ is a list of $n - k - 2$ pairs of integers drawn from $N \cup \{\epsilon\}$, where $\epsilon$ is an arbitrary number not in $N$. Dandelion codes can be sampled uniformly by a trivial linear-time

algorithm that uniformly chooses $k$ elements from $N$ to build $Q$, then uniformly samples $n - k - 2$ pairs of integers in $N \cup \{\epsilon\}$. Algorithm 1 contains a high-level description of our approach.

---

**Algorithm 1** Learning a structure of bounded treewidth by sampling Dandelion codes.

---

*% Takes a score function $s_i, i \in N$, and an integer $k$, and outputs a DAG $G^*$ of treewidth $\leq k$.*
1 Initialize $G^*$ as an empty DAG.
2 Repeat a certain number of iterations:
2.a Uniformly sample a Dandelion code $(Q, S)$ and decode it into $T_k$.
2.b Search for a DAG $G$ that maximizes the score function and is *compatible* with $T_k$.
2.c If $\sum_{i \in N} s_i(G_i) > \sum_{i \in N} s_i(G_i^*)$, update $G^*$.

---

We assume from now on that a $k$-tree $T_k$ is available, and consider the problem of searching for a compatible DAG that maximizes the score (Step 2.b). Korhonen and Parviainen [19] presented an algorithm (which we call K&P) that given an undirected graph $M$ finds a DAG $G$ maximizing the score function such that the moralization of $G$ is a subgraph of $M$. The algorithm runs in time and space $O(n)$ assuming the scores are part of the input (hence pre-computed and accessed at constant time). We can use their algorithm to find the optimal structure whose moral graph is a subgraph of $T_k$. We call this approach S+K&P to remind of ($k$-tree) sampling followed by K&P.

**Theorem 2.** *The size of the sampling space of S+K&P is less than $e^{n \log(nk)}$. Each of its iterations runs in linear time in $n$ (but exponential in $k$).*

According to the result above, the sampling space of S+K&P is not much bigger than that of standard order-based local search (which is approximately $e^{n \log n}$), especially if $k \ll n$. The practical drawback of this approach is the $\Theta(k3^k(k+1)!n)$ time taken by K&P to process each sampled $k$-tree, which forbids its use for moderately high treewidth bounds (say, $k \geq 10$). Our experiments in the next section further corroborate our claim: S+K&P often performs poorly even on small $k$, mostly due to the small number of $k$-trees sampled within the given time limit. A better approach is to sacrifice the optimality of the search for compatible DAGs in exchange of an efficiency gain. We next present a method based on sampling topological orderings that achieves such a goal.

Let $\mathcal{C}_i$ be the collection of maximal cliques of $T_k$ that contain a certain node $i$ (these can be obtained efficiently, as $T_k$ is chordal), and consider a topological ordering $<$ of $N$. Let $C_{<i} = \{j \in C : j < i\}$. We can find an optimal DAG $G$ compatible with $<$ and $T_k$ by making $G_i = \text{argmax}\{s_i(P) : P \subseteq C_{<i} : C \in \mathcal{C}_i\}$ for each $i \in N$. The graph $G$ is acyclic since each parent set $G_i$ respects the topological ordering by construction. Its treewidth is at most $k$ because both $i$ and $G_i$ belong to a clique $C$ of $T_k$, which implies that the moralization of $G$ is a subgraph of $T_k$.

Sampling topological orderings is both inefficient and wasteful, as different topological orderings impose the same constraints on the choices of $G_i$. To see this, consider the $k$-tree with edges 1–2,1–3,2–3,2–4 and 3–4. Since there is no edge connecting nodes 1 and 4 their relative ordering is irrelevant when choosing both $G_1$ or $G_4$. A better approach is to linearly order the nodes in each maximal clique.

A $k$-tree $T_k$ can be represented by a clique-tree structure, which comprises its maximal cliques $C_1, \ldots, C_{n+k-1}$ and a tree $T$ over the maximal cliques. Every two adjacent cliques in $T$ differ by exactly one node. Assume $T$ is rooted at a clique $R$, so we can unambiguously refer to the (single) parent of a (maximal) clique and to its children. A clique-tree structure as such can directly be obtained from the process of decoding a Dandelion code [6]. The procedure in Algorithm 2 shows how to efficiently obtain a collection of compatible orderings of the nodes of each clique of a $k$-tree.

---

**Algorithm 2** Sampling a partial order within a $k$-tree.

---

*% Takes a $k$-tree represented as a clique-tree structure $T$ rooted at $R$, and outputs a collection of orderings $\sigma_C$ for every maximal clique $C$ of $T$.*
1 Sample an order $\sigma_R$ of the nodes in $R$, paint $R$ black and the other maximal cliques white.
2 Repeat until all maximal cliques are painted black:
2.a Take a white clique $C$ whose parent clique $P$ in $T$ is black, and let $i$ be the single node in $C \setminus P$.
2.b Sample a relative order for $i$ with respect to $\sigma_P$ (i.e., insert $i$ into some arbitrary position of the projection of $\sigma_P$ onto $C$), and generate $\sigma_C$ accordingly; when done paint $C$ black.

---

Table 1: Number of variables in the data sets.

| nursery | breast | housing | adult | zoo | letter | mushroom | wdbc | audio | hill | community |
|---------|--------|---------|-------|-----|--------|----------|------|-------|------|-----------|
| 9 | 10 | 14 | 15 | 17 | 17 | 22 | 31 | 62 | 100 | 100 |

The cliques in Algorithm 2 are processed in topological ordering in the clique-tree structure, which ensures that the order $\sigma_P$ of the parent $P$ of a clique $C$ is already defined when processing $C$ (note that the order in which we process cliques does not restrict the possible orderings among nodes). At the end, we have a node ordering for each clique. Given such a collection of local orderings, we can efficiently learn the optimal parent set of every node $i$ by

$$G_i = \underset{P \subseteq C: P \sim \sigma_C, C \in \mathcal{C}_i}{\operatorname{argmax}} s_i(P), \tag{6}$$

where $P \sim \sigma_C$ denotes that the parent sets are constrained to be nodes smaller than $i$ in $\sigma_C$. In fact, the choices made in (6) can be implemented together with step 2.b of Algorithm 2, providing a slight increase of efficiency. We call the method obtained by Algorithm 1 with partial orderings established by Algorithm 2 and parent set selection by (6) as S2, in allusion to the double sampling scheme of $k$-trees and local node orderings.

**Theorem 3.** *S2 samples DAGs $\sigma$ on a sample space of size $k! \cdot (k+1)^{n-k}$, and runs in linear time in $n$ and $k$.*

The generation of partial orderings can also serve to implement the DAG search in S+K&P, by replacing the sampling with complete enumeration of them. Then Step 2.b would be performed for each compatible ordering $\sigma_P$ of the parent in a recursive way. Dynamic programming can be used to make the procedure more efficient. We have actually used this approach in our implementation of S+K&P. Finally, the sampling can be enhanced by some systematic search in the neighborhood of the sampled candidates. We have implemented and put in place a simple hill-climbing procedure for that, even though the quality of solutions does not considerably improve by doing so.

## 5 Experiments

We empirically analyzed the accuracy of the algorithms proposed here against each other and against the available implementations of TWILP (https://bitbucket.org/twilp/twilp/) and K&P (http://www.cs.helsinki.fi/u/jazkorho/aistats-2013/) on a collection of data sets from the UCI repository. The S+K&P and S2 algorithms were implemented (purely) in Matlab. The data sets were selected so as to span a wide range of dimensionality, and were preprocessed to have variables discretized over the median value when needed. Some columns of the original data sets *audio* and *community* were discarded: 7 variables of *audio* had a constant value, 5 variables of *community* have almost one different value per sample (such as personal data), and 22 variables had missing data (Table 1 shows the number of (binary) variables after pre-processing). In all experiments, we maximize the Bayesian Dirichlet equivalent uniform score with equivalent sample size equal to one.

### 5.1 Exact Solutions

We refer to our MILP formulation as simply MILP hereafter. We compared MILP, TWILP and K&P on the task of finding an optimal structure. Table 2 reports the running time on a selection of data sets of reasonably low dimensionality and small values for the treewidth bound. The experiments were run in a computer with 32 cores, memory limit of 64GB, time limit of 3h and maximum number of parents of three (the latter restriction facilitates the experiments and does not constrain the treewidth). On cases where MILP or TWILP did not finish we report also the error estimates from CPLEX (an error of $e\%$ means that the achieved solution is certainly not more than $e\%$ worse than the optimal). While we emphasize that one should be careful when directly comparing execution time between methods, as the implementations use different languages (we are running CPLEX 12.4, the original K&P uses a Cython compiled Python code, TWILP uses a Python interface to CPLEX to generate the cutting plane mechanism), we note that MILP goes much further in terms of which data sets and treewidth values it can compute. MILP has found the optimal structure in all instances, but was not able to certify its optimality in due time. TWILP found the optimum for

all treewidth bounds only on the nursery and breast data sets. The results also suggest that MILP becomes faster with the increase of the bound, while TWILP running times remain almost unaltered. This might be explained by the fact that the MILP formulation is complete and the increase of the bound facilitates encountering good solutions, while TWILP needs to generate constraints until an optimal solution can be certified.

Table 2: Time to learn an optimal Bayesian network subject to treewidth bound $w$. Dashes denote failure to solve due to excessive memory demand.

| method | $w$ | nursery $n=9$ | breast $n=10$ | housing $n=14$ | adult $n=15$ | mushroom $n=22$ |
|---|---|---|---|---|---|---|
| MILP | 2 | 1s | 31s | 3h [2.4%] | 3h [0.39%] | 3h [50%] |
| | 3 | <1s | 19s | 25m | 3h [0.04%] | 3h [19.3%] |
| | 4 | <1s | 8s | 80s | 40m | 3h [14.9%] |
| | 5 | <1s | 8s | 56s | 37s | 3h [11.2%] |
| TWILP | 2 | 5m | 3h [0.5%] | 3h [7%] | 3h [0.6%] | 3h [32%] |
| | 3 | 5s | 3h [3%] | 3h [9%] | 3h [0.7%] | 3h [31%] |
| | 4 | <1s | 3h [0.3%] | 3h [9%] | 3h [0.9%] | 3h [27%] |
| | 5 | <1s | 3h [0.5%] | 3h [7%] | 3h [0.9%] | 3h [23%] |
| K&P | 2 | 7s | 26s | 128m | 137m | – |
| | 3 | 72s | 5m | – | – | – |
| | 4 | 12m | 103m | – | – | – |
| | 5 | 131m | – | – | – | – |

## 5.2 Approximate Solutions

We used treewidth bounds of 4 and 10, and maximum parent set size of 3, except for *hill* and *community*, where it was set as 2 to help the integer programming approaches (which suffer the most from large parent sets). To be fair with all methods, we pre-computed scores, and considered them as input of the problem. Both MILP and TWILP used CPLEX 12.4 with a memory limit of 64GB to solve the optimizations. We have allowed CPLEX to run up to three hours, collecting the incumbent solution after 10 minutes. S+K&P and S2 have been given 10 minutes. This evaluation at 10 minutes is to be seen as an early-stage comparison for applications that need a reasonably fast response. To account for the intrinsic variability of the performance of the sampling methods with respect to the sampling seed, S+K&P and S2 were ran ten times on each data set with different seeds; we report the minimum, median and maximum obtained values over the runs.

Figure 1 shows the normalized scores (in percentage) of each method on each data set. The *normalized score* of a method that returns a solution with score $s$ on a certain data set is norm-score$(s) = (s - s_{empty})/(s_{max} - s_{empty})$, where $s_{empty}$ is the score of an empty DAG (used as baseline), and $s_{max}$ is the maximum score over all methods in that data set. Hence, a normalized score of 0 indicates the method found solutions as good as the empty graph (a trivial solution), whereas a normalized score of 1 indicates the method performed best on that data set.

The exponential dependence on treewidth of S+K&P prevents it to run with treewidth bound greater than 6. We see from the plot on the left that S2 is largely superior to S+K&P, even though the former finds suboptimal networks for each given $k$-tree. This suggests that finding good $k$-trees is more important than selecting good networks for a given $k$-tree. We also see that both integer programming formulations scale poorly with the number of variables, being unable to obtain satisfactory solutions for data sets with more than 50 variables. For the *hill* data set and treewidth $\leq 4$, MILP was not able to find a feasible solution within 10 minutes, and could only find the trivial solution (empty DAG) after 3 hours; TWILP did not find any solution even after 3 hours. On the *community* data set with treewidth $\leq 4$, neither MILP nor TWILP found a solution within 3 hours. For treewidth $\leq 10$ the integer programming approaches performed even worse: TWILP could not provide a solution for the audio, hill and community data sets; MILP could only find the empty graph.

Since both S+K&P and S2 were implemented in Matlab, the comparison with either MILP or TWILP within the same time period (10 minutes) might be unfair (one could also try to improve the MILP formulation, although it will eventually suffer from the problems discussed in Section 3). Nevertheless, the results show that S2 is very competitive even under implementation disadvantage.

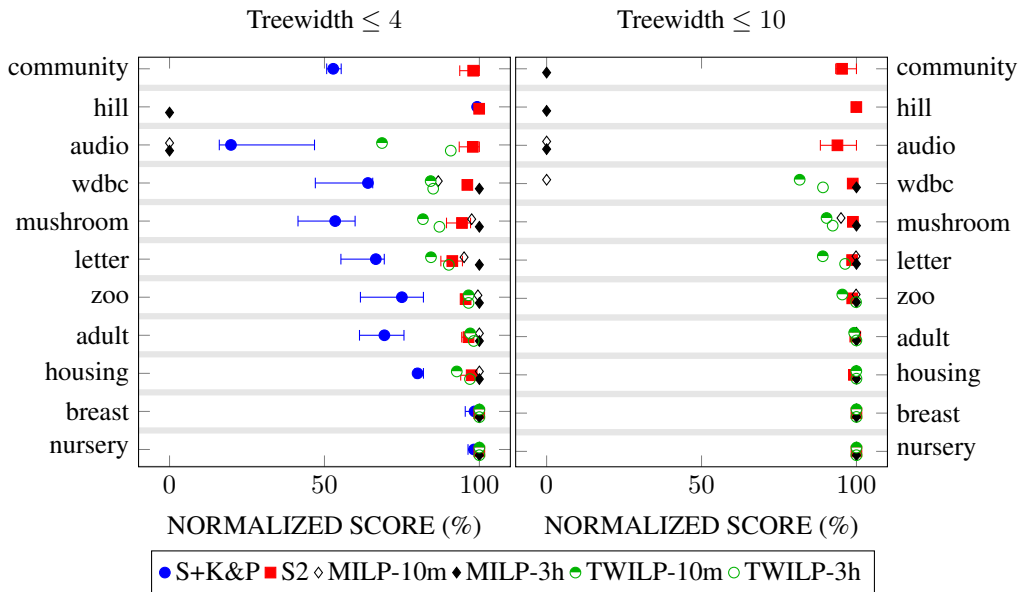

Figure 1: Normalized scores. Missing results indicate failure to provide a solution.

## 6 Conclusions

We presented exact and approximate procedures to learn Bayesian networks of bounded treewidth. The exact procedure is based on a MILP formulation, and is shown to outperform other methods for exact learning, including the different MILP formulation proposed in [21]. Our MILP approach is also competitive when used to produce approximate solutions. However, due to the cubic number of constraints, the MILP formulation cannot cope with very large domains, and there is probably little we can do to considerably improve this situation. Constraint generation techniques [3] are yet to be explored, even though we do not expect them to produce dramatic performance gains – the competing objectives of maximizing score and bounding treewidth usually lead to the generation of a large number of constraints.

To tackle large problems, we developed an approximate algorithm that samples $k$-trees and then searches for compatible structures. We derived two variants by trading off the computational effort spent in sampling $k$-trees and in searching for compatible structures. The sampling-based methods are empirically shown to provide fairly accurate solutions and to scale to large domains.

## Acknowledgments

We thank the authors of [19, 21] for making their software publicly available and the anonymous reviewers for their useful suggestions. Most of this work has been performed while C. P. de Campos was with the Dalle Molle Institute for Artificial Intelligence. This work has been partially supported by the Swiss NSF grant 200021_146606/1, by the São Paulo Research Foundation (FAPESP) grant 2013/23197-4, and by the grant N00014-12-1-0868 from the US Office of Navy Research.

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
