[Supplementary Material · supp.pdf]

# Advances in Learning Bayesian Networks of Bounded Treewidth: Supplementary Material

Siqi Nie
Rensselaer Polytechnic Institute
Troy, NY, USA
nies@rpi.edu

Denis D. Mauá
University of São Paulo
São Paulo, Brazil
denis.maua@usp.br

Cassio P. de Campos
Queen's University Belfast
Belfast, UK
c.decampos@qub.ac.uk

Qiang Ji
Rensselaer Polytechnic Institute
Troy, NY, USA
qji@ecse.rpi.edu

## 1   Introduction

This document contains additional content such as explanations, proofs and tables that had to be removed from the main paper to meet the space requirements. The goal of this work is to develop methods that search for a DAG structure $G^*$ such that

$$G^* = \arg \max_{G \in \mathcal{G}_{N,k}} \sum_{i \in N} s_i(G_i) \,, \tag{1}$$

where $\mathcal{G}_{N,k}$ is the set of all DAGs with node set $N$ and treewidth at most $k$.

## 2   Graph-Theoretic Concepts

We say that a cycle in an undirected graph has a chord if there are two nodes in the cycle which are connected by an edge outside the cycle. A chordal graph is an undirected graph in which all cycles of length four or more have a chord. Any graph can be made chordal by inserting edges, a process called *chordalization* [1, 3]. The treewidth of a chordal graph is the size of its largest clique (a fully connected subgraph) minus one. The treewidth of an arbitrary undirected graph is the minimum treewidth over all chordalizations of it. The *moral graph* of a DAG is the undirected graph obtained by connecting any two nodes with a common child and dropping arc directions, a process known as *moralization*. The treewidth of a DAG is the treewidth of its corresponding moral graph. The treewidth of a Bayesian network is the treewidth of its underlying DAG.

An *elimination order* is a linear ordering of the nodes in a graph. We say that an elimination order is *perfect* if for every node in the order its higher-ordered neighbors form a clique (i.e., are pairwise connected). A graph admits a perfect elimination order if and only if it is chordal. Perfect elimination orders can be computed in linear time if they exist. The *elimination* of a node according to an elimination order is the process of pairwise connecting all of its higher-ordered neighbors. Thus, the elimination of all nodes produces a chordal graph for which the elimination order used is perfect. The edges inserted by the elimination process are called *fill-in* edges. Given a perfect elimination order, the treewidth of the graph can be computed as the maximum number of higher ordered neighbors in the graph.

# 3 Omitted Proofs

## 3.1 MILP Approach

Maximize:

$$\sum_{it} p_{it} \cdot s_i(P_{it}) \tag{2}$$

Subject to:

$$\sum_{j \in N} y_{ij} \leq w, \qquad\qquad \forall i \in N, \tag{3a}$$

$$(n+1) \cdot y_{ij} \leq n + z_j - z_i, \qquad\qquad \forall i,j \in N, \tag{3b}$$

$$y_{ij} + y_{ik} - y_{jk} - y_{kj} \leq 1, \qquad\qquad \forall i,j,k \in N, \tag{3c}$$

$$\sum_t p_{it} = 1, \qquad\qquad \forall i \in N, \tag{4a}$$

$$(n+1)p_{it} \leq n + v_j - v_i, \qquad\qquad \forall i \in N, \forall t \in \{1,\ldots,r_i\}, \forall j \in P_{it}, \tag{4b}$$

$$p_{it} \leq y_{ij} + y_{ji}, \qquad\qquad \forall i \in N, \forall t \in \{1,\ldots,r_i\}, \forall j \in P_{it}, \tag{4c}$$

$$p_{it} \leq y_{jk} + y_{kj}, \qquad\qquad \forall i \in N, \forall t \in \{1,\ldots,r_i\}, \forall j,k \in P_{it}, \tag{4d}$$

$$z_i \in [0,n],\ v_i \in [0,n],\ y_{ij} \in \{0,1\},\ p_{it} \in \{0,1\} \qquad \forall i,j \in N, \forall t \in \{1,\ldots,r_i\}. \tag{5}$$

**Lemma 1.** *Let $z_i, y_{ij}$, $i,j \in N$, be variables satisfying Constraints* (4) *and* (5)*. Then the undirected graph $M = (N, E)$, where $E = \{ij \in N \times N : y_{ij} = 1 \text{ or } y_{ji} = 1\}$, is chordal and has treewidth at most $w$. Moreover, any elimination order that extends the partial order induced by $z_i$ is perfect for $M$.*

*Proof.* The variables $z_i$, $i \in N$, partially define an elimination order of the nodes: a node $i$ is eliminated before node $j$ if $z_i < z_j$ (the specification is partial since its allows for two nodes $i$ and $j$ with $z_i = z_j$). This order need not be linear because there are cases where multiple linearizations of the partial order are equally good in building a chordalization (i.e., in minimizing the maximum clique size of $M$). In such cases, two nodes $i$ and $j$ might be assigned the same value $z_i = z_j$ indicating that eliminating $z_i$ before $z_j$ or the converse results in chordal graphs of the same treewidth. The variables $y_{ij}$, $i,j \in N$ denote whether node $i$ precedes $j$ in the order (i.e., whether $z_i < z_j$) *and* an edge exists among them in $M$ (recall that an elimination process always produces a chordal graph). Constraint (3a) ensures $M$ has treewidth at most $w$ by bounding the number of higher-ordered neighbors of every node $i$ (which is an alternative way of defining the treewidth of chordal graphs). Constraint (3b) allows $y_{ij}$ to be 1 only if $j$ appears after $i$ in the order (it in fact requires that $z_j \geq z_i + 1$ to allow $y_{ij}$ to be one). Constraint (3c) guarantees that any elimination ordering induced by $z_i$, $i \in N$, is perfect for $M$: if $j$ and $k$ are higher ordered neighbors of $i$ in $M$, then $j$ and $k$ are also neighbors in $M$, that is, either $y_{jk}$ or $y_{kj}$ must be 1. $\qquad\square$

**Lemma 2.** *Let $v_i, p_{it}$, $i \in N, t = 1, \ldots, r_i$, be variables satisfying Constraints* (4) *and* (5)*. Then the directed graph $G = (N, A)$, where $G_i = \{j : p_{it} = 1 \text{ and } j \in P_{it}\}$, is acyclic and valid. Moreover the moral graph of $G$ is a subgraph of the graph $M$ defined in the previous lemma.*

*Proof.* The variables $v_i$, $i \in N$ partially specify a topological order of the nodes in $G$: if $v_i > v_j$ then $j$ is not an ancestor of $i$. The variables $p_{it}$, $i \in N$, $t = 1, \ldots, r_i$, are represent whether the $t$-th parent set $P_{it}$ in $\mathcal{P}_i$ was chosen for node $i$. Constraint (4a) enforces that exactly one parent set is chosen for each node. Constraint (4b) forces those parent set choices to be acyclic, that is, to respect the topological order induced by the variables $v_i$ (with ties broken arbitrarily for nodes $i, j$ with $v_i = v_j$). That order need not be linear, as only the relative orders of nodes that are connected in $M$ are relevant because Constraints (4c) and (4d) ensure that arcs appear in $G$ only if the corresponding edges in the moral graph of $G$ exist in $M$ (Constraint (4d) is responsible for having the moralization of the graph falling inside $M$). $\qquad\square$

**Theorem 1.** *Any solution to the MILP can be decoded into a valid DAG of treewidth less than $w$. In particular, the decoding of an optimum solution solves* (1)*.*

*Proof.* According to Lemma 2, any solution can be decoded into a valid DAG $G$ whose moral graph is a subgraph of the $M$, which by Lemma 1 has treewidth at most $w$. If the solution is optimal, the corresponding DAG $G$ maximizes the score function by construction (as it maximizes (2)), hence solving (1). $\qquad\square$

## 3.2 Sampling Based Approach

**Theorem 2.** *The sampling space of S+k&P is less than $e^{n \log(nk)}$. Each of its iterations runs in linear time in $n$ (but exponential in $k$).*

*Proof.* The follow equality holds [2]:

$$|\mathcal{T}_{n,k}| = \binom{n}{k}(k(n-k)+1)^{n-k-2} .$$

It is not hard to see that the maximum happens for $k \leq n/2$ (because of the symmetry of $\binom{n}{k}$ and of $k(n-k)$ around $n/2$, while $n-k-2$ decreases with the increase of $k$). By manipulating this number and applying Stirling's approximation for the factorials, we obtain:

$$|\mathcal{T}_{n,k}| \leq \frac{\sqrt{n}e^{n \log n + 1 - n}}{\left(\frac{n-k}{e}\right)^{n-k}\left(\frac{k}{e}\right)^{k}}k^{n-k-2}(n-k)^{n-k-2}$$

$$\leq \frac{e\sqrt{n}}{(n-k)^2}e^{n \log n}k^{n-2k-2} \leq e^{n \log n + (n-2k) \log k} ,$$

which is less than $e^{n \log(nk)}$. The decoding algorithm has complexity linear in $n$, as well as the method to uniformly sample a Dandelion code and the method to find the best DAG consistent with a $k$-tree. $\square$

**Theorem 3.** *S2 samples DAGs $\sigma$ on a sample space of size $k! \cdot (k+1)^{n-k}$, and runs in linear time in $n$ and $k$.*

*Proof.* The sampling of the $k+1$ nodes in the root clique takes time $O(k)$ by sampling one of the $(k+1)!$ ways to permute the nodes. Step 2a can be done in constant time if the clique-tree structure is implemented as a dictionary. For each iteration of Step 2c, we take $O(k)$ time to decide on one of the $k+1$ possible relative orderings for node $i$ (a position between $0$ and $k$) and produce a sorted array of nodes (assuming partial orders are thus represented). Since the clique-tree has $n+k-1$ nodes, the total time is $O(k \cdot n)$ and the sampling space is $(k+1)! \cdot (k+1)^{n-k-1} = k! \cdot (k+1)^{n-k}$. $\square$

# 4 Experiments

Table 1 contains more details about the data sets used in the experiments in the paper.

Table 1: Dimensions of data sets.

| DATASET | VAR. | SAMPLES |
|---|---|---|
| nursery | 9 | 12960 |
| breast | 10 | 699 |
| housing | 14 | 506 |
| adult | 15 | 32561 |
| zoo | 17 | 101 |
| letter | 17 | 20000 |
| mushroom | 22 | 8124 |
| wdbc | 31 | 569 |
| audio | 62 | 200 |
| hill | 100 | 606 |
| community | 100 | 1994 |

Figure 1 reproduces Figure 1 of the paper at a higher scale. Tables 2–5 show the values used to produce the plots in the Figure.

Table 2: Performance of integer programming methods with **treewidth limit of 4**. Values have been subtracted from the minimum score (i.e., the empty DAG score). Maximum, median and minimum are taken with respect to 10 runs with different seeds. A symbol – indicates the method was not able to produce a solution within the time limit.

| DATASET | MILP$^{10m}$ | TWILP$^{10m}$ | $\frac{\text{MILP}^{10m}-\text{TWILP}^{10m}}{\text{MILP}^{10m}-\text{EMPTY}}$ | MILP$^{3h}$ | TWILP$^{3h}$ | $\frac{\text{MILP}^{3h}-\text{TWILP}^{3h}}{\text{MILP}^{3h}-\text{EMPTY}}$ |
|---|---|---|---|---|---|---|
| nursery | $4.5986\cdot10^3$ | $4.5986\cdot10^3$ | 0.000001 | $4.5986\cdot10^3$ | $4.5986\cdot10^3$ | 0.000001 |
| breast | $1.8692\cdot10^3$ | $1.8692\cdot10^3$ | 0.000004 | $1.8692\cdot10^3$ | $1.8692\cdot10^3$ | 0.000004 |
| housing | $1.5030\cdot10^3$ | $1.3934\cdot10^3$ | 0.072919 | $1.5030\cdot10^3$ | $1.4573\cdot10^3$ | 0.030393 |
| adult | $1.4949\cdot10^4$ | $1.4508\cdot10^4$ | 0.029517 | $1.4949\cdot10^4$ | $1.4674\cdot10^4$ | 0.018381 |
| zoo | $4.5319\cdot10^2$ | $4.3946\cdot10^2$ | 0.030292 | $4.5533\cdot10^2$ | $4.3946\cdot10^2$ | 0.034850 |
| letter | $3.9106\cdot10^4$ | $3.4710\cdot10^4$ | 0.112419 | $4.1134\cdot10^4$ | $3.7078\cdot10^4$ | 0.098605 |
| mushroom | $4.5241\cdot10^4$ | $3.7941\cdot10^4$ | 0.161368 | $4.6391\cdot10^4$ | $4.0413\cdot10^4$ | 0.128860 |
| wdbc | $4.6128\cdot10^3$ | $4.4821\cdot10^3$ | 0.028352 | $5.3182\cdot10^3$ | $4.5254\cdot10^3$ | 0.149077 |
| audio | $0.0000\cdot10^0$ | $2.8728\cdot10^2$ | $-\infty$ | $0.0000\cdot10^0$ | $3.8026\cdot10^2$ | $-\infty$ |
| hill | – | – | – | $0.0000\cdot10^0$ | – | – |
| community | – | – | – | – | – | – |

Table 3: Performance of sampling methods within 10 minutes of time limit and **treewidth limit of 4**. Values have been subtracted from the minimum score (i.e., the empty DAG score). Maximum, median and minimum are taken with respect to 10 runs with different seeds. A symbol – indicates the method was not able to produce a solution within the time limit.

| DATASET | S+K&P$^{min}$ | S+K&P$^{median}$ | S+K&P$^{max}$ | S2$^{min}$ | S2$^{median}$ | S2$^{max}$ |
|---|---|---|---|---|---|---|
| nursery | $4.4303\cdot10^3$ | $4.5185\cdot10^3$ | $4.5222\cdot10^3$ | $4.5986\cdot10^3$ | $4.5986\cdot10^3$ | $4.5986\cdot10^3$ |
| breast | $1.7840\cdot10^3$ | $1.8400\cdot10^3$ | $1.8578\cdot10^3$ | $1.8675\cdot10^3$ | $1.8684\cdot10^3$ | $1.8692\cdot10^3$ |
| housing | $1.1912\cdot10^3$ | $1.2032\cdot10^3$ | $1.2317\cdot10^3$ | $1.4125\cdot10^3$ | $1.4657\cdot10^3$ | $1.4751\cdot10^3$ |
| adult | $9.1653\cdot10^3$ | $1.0367\cdot10^4$ | $1.1311\cdot10^4$ | $1.4097\cdot10^4$ | $1.4425\cdot10^4$ | $1.4604\cdot10^4$ |
| zoo | $2.8054\cdot10^2$ | $3.4147\cdot10^2$ | $3.7312\cdot10^2$ | $4.2878\cdot10^2$ | $4.3510\cdot10^2$ | $4.4109\cdot10^2$ |
| letter | $2.2752\cdot10^4$ | $2.7387\cdot10^4$ | $2.8510\cdot10^4$ | $3.6022\cdot10^4$ | $3.7541\cdot10^4$ | $3.8888\cdot10^4$ |
| mushroom | $1.9236\cdot10^4$ | $2.4802\cdot10^4$ | $2.7787\cdot10^4$ | $4.1472\cdot10^4$ | $4.3787\cdot10^4$ | $4.5070\cdot10^4$ |
| wdbc | $2.5026\cdot10^3$ | $3.4056\cdot10^3$ | $3.4878\cdot10^3$ | $5.0381\cdot10^3$ | $5.1117\cdot10^3$ | $5.1548\cdot10^3$ |
| audio | $6.7275\cdot10^1$ | $8.3067\cdot10^1$ | $1.9602\cdot10^2$ | $3.9170\cdot10^2$ | $4.1005\cdot10^2$ | $4.1892\cdot10^2$ |
| hill | $4.0837\cdot10^4$ | $4.0936\cdot10^4$ | $4.1073\cdot10^4$ | $4.1197\cdot10^4$ | $4.1211\cdot10^4$ | $4.1236\cdot10^4$ |
| community | $2.2018\cdot10^4$ | $2.2935\cdot10^4$ | $2.4053\cdot10^4$ | $4.0662\cdot10^4$ | $4.2544\cdot10^4$ | $4.3411\cdot10^4$ |

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

Table 4: Performance of integer programming methods with **treewidth limit of 10**. Values have been subtracted from the minimum score (i.e., the empty DAG score). Maximum, median and minimum are taken with respect to 10 runs with different seeds. A symbol – indicates the method was not able to produce a solution within the time limit.

| DATASET | MILP$^{10m}$ | TWILP$^{10m}$ | $\frac{\text{MILP}^{10m}-\text{TWILP}^{10m}}{\text{MILP}^{10m}-\text{EMPTY}}$ | MILP$^{3h}$ | TWILP$^{3h}$ | $\frac{\text{MILP}^{3h}-\text{TWILP}^{3h}}{\text{MILP}^{3h}-\text{EMPTY}}$ |
|---|---|---|---|---|---|---|
| nursery | $4.5986 \cdot 10^3$ | $4.5986 \cdot 10^3$ | 0.000001 | $4.5986 \cdot 10^3$ | $4.5986 \cdot 10^3$ | 0.000001 |
| breast | $1.8692 \cdot 10^3$ | $1.8692 \cdot 10^3$ | 0.000004 | $1.8692 \cdot 10^3$ | $1.8692 \cdot 10^3$ | 0.000004 |
| housing | $1.5030 \cdot 10^3$ | $1.5022 \cdot 10^3$ | 0.000499 | $1.5030 \cdot 10^3$ | $1.5030 \cdot 10^3$ | 0.000005 |
| adult | $1.5013 \cdot 10^4$ | $1.4902 \cdot 10^4$ | 0.007376 | $1.5013 \cdot 10^4$ | $1.5009 \cdot 10^4$ | 0.000222 |
| zoo | $4.5560 \cdot 10^2$ | $4.3528 \cdot 10^2$ | 0.044598 | $4.5614 \cdot 10^2$ | $4.5554 \cdot 10^2$ | 0.001308 |
| letter | $4.1411 \cdot 10^4$ | $3.6960 \cdot 10^4$ | 0.107495 | $4.1469 \cdot 10^4$ | $3.9949 \cdot 10^4$ | 0.036655 |
| mushroom | $4.6230 \cdot 10^4$ | $4.3962 \cdot 10^4$ | 0.049069 | $4.8656 \cdot 10^4$ | $4.4924 \cdot 10^4$ | 0.076695 |
| wdbc | $1.0000 \cdot 10^{-2}$ | $4.4262 \cdot 10^3$ | $-442,615.000000$ | $5.4195 \cdot 10^3$ | $4.8336 \cdot 10^3$ | 0.108100 |
| audio | $0.0000 \cdot 10^0$ | – | – | $0.0000 \cdot 10^0$ | – | – |
| hill | – | – | – | $0.0000 \cdot 10^0$ | – | – |
| community | – | – | – | $0.0000 \cdot 10^0$ | – | – |

Table 5: Performance of sampling methods within 10 minutes of time limit and **treewidth limit of 10**. Values have been subtracted from the minimum score (i.e., the empty DAG score). Maximum, median and minimum are taken with respect to 10 runs with different seeds. A symbol – indicates the method was not able to produce a solution within the time limit.

| DATASET | S2$^{\min}$ | S2$^{\text{median}}$ | S2$^{\max}$ |
|---|---|---|---|
| nursery | $4.5986 \cdot 10^3$ | $4.5986 \cdot 10^3$ | $4.5986 \cdot 10^3$ |
| breast | $1.8692 \cdot 10^3$ | $1.8692 \cdot 10^3$ | $1.8692 \cdot 10^3$ |
| housing | $1.4814 \cdot 10^3$ | $1.4905 \cdot 10^3$ | $1.5024 \cdot 10^3$ |
| adult | $1.4862 \cdot 10^4$ | $1.4955 \cdot 10^4$ | $1.4976 \cdot 10^4$ |
| zoo | $4.4555 \cdot 10^2$ | $4.4991 \cdot 10^2$ | $4.5343 \cdot 10^2$ |
| letter | $4.0601 \cdot 10^4$ | $4.0889 \cdot 10^4$ | $4.1364 \cdot 10^4$ |
| mushroom | $4.7268 \cdot 10^4$ | $4.8102 \cdot 10^4$ | $4.8614 \cdot 10^4$ |
| wdbc | $5.3285 \cdot 10^3$ | $5.3545 \cdot 10^3$ | $5.3691 \cdot 10^3$ |
| audio | $3.9449 \cdot 10^2$ | $4.1907 \cdot 10^2$ | $4.4649 \cdot 10^2$ |
| hill | $4.1238 \cdot 10^4$ | $4.1349 \cdot 10^4$ | $4.1368 \cdot 10^4$ |
| community | $4.3188 \cdot 10^4$ | $4.4135 \cdot 10^4$ | $4.6289 \cdot 10^4$ |

Figure 1: Normalized scores. Missing results indicate failure to provide a solution.