[Reviews · NeurIPS 2014]

Submitted by Assigned_Reviewer_10

This paper presents two very natural methods for searching for Bayesian networks of treewidth <= k. The exact method uses a MILP to search jointly over (Bayesian network, elimination order, chordal graph induced by elimination on the moralized network). The approximate method repeatedly samples a random k-tree and then uses dynamic programming or sampling to find the best network whose moralization is a subgraph of the k-tree (any network of treewidth <= k can be obtained in this way).

The paper is clear and the methods seem like textbook material. Unfortunately, I haven't had time to go read the closely related prior work cited at lines 051-058 or 068-071, so I can't comment on the size of the incremental contribution. If none of the reviewers are familiar with those papers, then I think one of us should go look at them.

Algorithm 1 (lines 236-244) does independent sampling of k-trees. Would search by hill-climbing or simulated-annealing be more efficient, or a genetic algorithm? It seems that such algorithms could be built in a similar way.

It is not obvious what distribution is sampled from by Algorithm 2 (lines 292-300). Line 073 says it is uniform over the set of partial orders consistent with the k-tree, but this is not obvious. In particular, the input clique tree is assumed to be rooted (i.e., the choice of root clique is NOT sampled), so doesn't that mean that some orders are impossible?

In general, I wish some effort were made to use the local score functions to influence the sampling distributions in the methods of section 4. Presumably this could provide a speedup over uniform sampling.

The experiments with the approximate methods could be presented more clearly. All of these algorithms are anytime algorithms, so why not show the score (with error bars) as a function of time?

Other than that, experimental results and discussion generally look ok. I notice one anomaly: why does the old method TWILP-10m strongly beat the new method MILP-10m on the wdbc dataset with treewidth >= 10?

FURTHER COMMENTS TO AUTHORS

line 220: You take a (k+1)-clique as the base case of a k-tree, but [7] takes a k-clique as the base case. (I may prefer your definition but I note the discrepancy.)

line 083: G = {N,A} should better be G = (N,A) or more standardly G = (V,E).

lines 160-173: Please provide brief interpretations of the variables and constraints. I believe that the p_{it} variables define the Bayesian network with the v_i specifying a variable ordering that is respected by the edge direction. The y_{ij} variables specify a triangulated moralized version of this network (plus optional extra edges). More precisely, the triangulation is obtained by at least one elimination ordering, and the direction of the y_{ij} edges respect this ordering, with the z_i certifying that the y_{ij} edges form a DAG. You should gloss the constraints with comments to help the reader: (3a) ensures that the induced width is at most w, (3b) says "if y_{ij}=1 then z_j >= z_i + 1", (3c) says "y_{ij} ^ y_{ik} -> y_{jk} v y_{kj}" which creates an induced edge between j and k when eliminating i, (4a) enforces that each variable chooses exactly one of its allowed parent sets, and so on, with (4d) enforcing moralization.

line 178: "the partial order induced by z_i": no, I think the partial order should be regarded as the transitive closure of the y_{ij}. That is what determines the set of possible elimination orders that are consistent with this solution, i.e., the possible orders are the topological sorts of the DAG formed by the y_{ij}. The z_i serve to certify that the y_{ij} do form a DAG, but they may commit to a stronger ordering than necessary, since two vertices that are unordered according to the y graph will not necessarily tie in their z values. (The z values are weakly ordered (i.e., totally ordered except for ties), but the particular choice of weak ordering is not determined by the y graph and is unimportant.)

line 215: (ii) should be "second".
lines 228, 231: typos for "Dandelion." Also, from [17] (section 4), it seems these should actually be called "generalized Dandelion codes," based on earlier work called Dandelion codes.
line 226: please clarify whether these are k-trees OVER A GIVEN SET OF VERTICES, which is what you need for your application, in which the vertices denote given variables. The definition of k-trees in the previous paragraph adds new vertices as needed, and one could think that "sampling a k-tree" means sampling from the space of such graphs up to isomorphism.

Why does section 3 use w for the treewidth while section 4 uses k for the treewidth?

line 249: you say O(n); what is n?
line 252: please give a parametric big-O complexity. E.g., you say exponential but you don't give a formula . So the reader has to think too hard to figure out what the base of the exponential is, whether this exponential is multiplied by n or added to n, etc. You do give something later on at line 258, but shouldn't the size of the parent sets (see line 154) figure into the formula?
line 339: clarify "error estimates." Does 50% that the current upper bound on model score was 1.5 times the model score of the current best solution?

Summary: Good and usable algorithms for an important problem. Clearly explained for the most part. Experiments are a reasonable demonstration of the proposed methods.
Before accepting, should check how much has been added here to previous work.

Submitted by Assigned_Reviewer_12

Exact and approximate methods for learning BNs of bounded treewidth
are presented and evaluated. The paper is clearly written and
addresses a worthwhile problem.

The exact approach is the MILP presented in Section 3. It rules out
cycles using the approach given by Cussens et al. The treewidth
constraint is encoded by requiring that the number of neighbours at
elimination time is at most the treewidth. In both cases "big M"
constraints are used.

The authors are happy that their MILP approach does not require
constraint generation. It is true that this makes life simpler (one
does not need to write a constraint generation algorithm - and perhaps
the solver is generating some nice ones for you). On the other hand
big M constraints typically lead to a loose linear relaxation which
will lead to a decrease in performance - constraint generation
approaches are chosen precisely to address this problem. Also since
constraint generation is ruled out this naturally leads to a limit on
size - we have a cubic number of constraints. This gets us into
trouble pretty soon if we insist on posting all such constraints from
the start.

On the other hand the presented exact approach performs well when compared
to TWILP which *does* use constraint generation. But, as the authors
state, one should be careful with the empirical comparisons due to
differences in implementation/language. It would have been more useful
to have a comparison of TWILP and MILP in terms of things like nodes
expanded and how many cutting planes CPLEX generated.

I'm not convinced by the comment that constraint generation techniques
"seem less effective" here due to the treewidth constraint. As noted
in Section 3 checking treewidth is NP-hard, but here the issue is
which of the possibly many constraints 3c and 4d to post at any point
in solving - why should that be so difficult.

I do not agree that order-based local search where orders are sampled
is "one of the most successful approaches for learning Bayesian
networks" - in my experience it's pretty useless. Nonetheless the
approximate approach presented here does better (on big problems) than
the two MIP exact approaches.

Although this paper raised a lot of questions I would have liked
answered, it does offer nicely straightforward approaches to the
problem which perform reasonably well in practice. I judge this to be
a sufficient contribution.

Summary: The exact and approximate approaches presented here are fairly well-motivated and perform well.

Submitted by Assigned_Reviewer_29

The work presents novel exact and approximate algorithms for learning Bayesian networks of bounded treewidth. The exact method combines mixed integer linear programming formulations for structure learning and treewidth computation. The approximate method consists in sampling k-trees and subsequently selecting the best structure whose moral graph is a subgraph of that k-tree. Empirical evaluation demonstrates the advantageous of the methods over other state-of-the-art methods.

This is a high quality paper. The paper is clearly written, the addressed problem is important, the presented methods are novel and the solution they provide is shown to be better than previous work.

Summary: This is a high quality paper. The paper is clearly written, the addressed problem is important, the presented methods are novel and the solution they provide is shown to be better than previous work.
Author Feedback
Author rebuttal: We thank all reviewers for the thoughtful comments and suggestions. They will be taken into consideration to improve the presentation of the paper. All minor issues will be certainly fixed, and for brevity, here we will comment on main points only.

Assigned_Reviewer_10:

As suggested, other searches, such as hill-climbing or simulated-annealing, can be used too. In fact, we have experimented with hill-climbing, and combining it with order-based sampling slightly improves results (hill-climbing alone did slightly worse). We will point this out.

We shall clarify in the paper the reason why Algorithm 2 samples over all partial orders consistent with the given k-tree (that is, all manners of directing arcs of the k-tree without creating cycles). The 'rooted' tree does not interfere with that, the root is only used to schedule the algorithm steps, but we can still consider all arc directions. We will clarify this matter.

Comparing TWILP and MILP at 10 minutes is to be seen as an early-stage comparison regarding their searches. TWILP superiority might be related to some cutting planes that were effective for that particular problem instance. In fact, our experiments showed us that MILP was superior in most cases, but it cannot be taken as being always superior (e.g. TWILP is superior to MILP also for dataset audio with treewidth 4). Further investigation on the practical usefulness of cutting planes is a possible future work (in fact, this topic is already under investigation, see e.g. ref. [19]). We will point that out too.

Assigned_Reviewer_12:

While we have seen good performance without constraint generation in our experiments, constraint generation techniques might be useful here, and so we will tone down our sentence about it. We meant that their application is not so obvious, because maximising score and bounding treewidth are competing tasks. Scenarios with _global_ competing objectives usually yield many constraints very quickly (we have seen that happening in our experiments). However, we agree this is to be further explored and so we will rephrase the sentence.

We have obtained very good empirical results with order-based sampling to support our excitement about it, but we agree that our sentence is too strong, so we will rephrase it.

Assigned_Reviewer_29:

We appreciate the very positive comments from the reviewer.